

# Restored freshwater flow and estuarine benthic communities in the northern Gulf of Mexico: research trends and future needs

Jillian C. Tupitza and Cassandra N. Glaspie

Department of Oceanography and Coastal Sciences, Louisiana State University and Agricultural and Mechanical College, Baton Rouge, LA, United States of America

## ABSTRACT

Restoring river connectivity to rebuild and sustain land is a promising restoration strategy in coastal areas experiencing rapid land loss, such as the Mississippi river delta. Results of these large-scale hydrologic changes are preliminary, and there exists limited empirical evidence regarding how benthic communities will respond, specifically in Barataria Bay and Breton Sound in southeast Louisiana. In this review, the body of existing research in this geographic region pertaining to the drivers of benthic community response that are related to restored freshwater flow and sediment deposition is examined. Overall trends include (1) potential displacement of some species down-estuary due to reduced salinities; (2) temporary lower diversity in areas closest to the inflow; (3) increased benthic production along the marsh edge, and in tidal bayous, as a result of nutrient loading; (4) more habitat coverage in the form of submerged aquatic vegetation; and (5) reduced predation pressure from large and/or salinity-restricted predators. These trends highlight opportunities for future research that should be conducted before large-scale hydrologic changes take place.

## INTRODUCTION

The rate of relative sea level rise (RSLR) along the Mississippi Delta approaches 10 mm $yr^{-1}$ in some areas due to eustatic sea level rise (ESLR) exacerbated by land subsidence, construction of oil and gas canals, and over a century of hydrologic disconnect in the form of containment levees (*Snedden et al., 2007*; *Martin, 2002*; *Turner & Boyer, 1997*). Dynamic wetlands that have been disconnected from Mississippi River sediments since the 20th century are unable to keep pace with RSLR because they are bounded by coastline urbanization, and are simultaneously eroded by storms and human activities (*Day et al., 2009*; *Wheelock, 2003*; *Snedden et al., 2007*). The conservative rate of land loss in these areas is 26.7 $km^2$ $yr^{-1}$ (*US Army Corps of Engineers, 2004*). A long-term restoration effort has been undertaken along the Louisiana coastline, including proposed construction of sediment diversions designed to deliver mineral sediments carried by freshwater from the

Corresponding author
Jillian C. Tupitza, jtupit1@lsu.edu

Mississippi River to the surrounding marshlands of Barataria Bay and Breton Sound (*CPRA, 2017*; *US Army Corps of Engineers, 2004*; *Martin, 2002*). The main goal of this project is to manage encroaching RSLR by building land, but the result will likely precipitate many other changes to the ecosystem.

The first sediment diversion is proposed at Mile 61 of the Mississippi, in the Plaquemines Parish, LA, and will divert sediment into Barataria Bay (*CPRA, 2017*; Fig. 1). Barataria Bay is a diurnal, well-mixed estuary of 167,300 ha located just west of the Mississippi River (*La Peyre & Birdsong, 2008*; Fig. 1). This shallow bay consists of a network of smaller bays, and is separated from the Gulf of Mexico by a number of barrier islands (*La Peyre & Birdsong, 2008*). The second sediment diversion is proposed at Mile 68 of the Mississippi, in Bertrandville, LA, and will divert sediment into Breton Sound (*CPRA, 2017*; Fig. 1). Breton Sound is a 271,000 ha estuary located just to the east of the Mississippi River (Fig. 1). This bathymetrically complex estuary is semi-enclosed, and is bounded by levees to the south and west (*Wang et al., 2017*). The mid-Barataria and mid-Breton sediment diversions, when constructed, will each be capable of diverting 2,124 $m^3 s^{-1}$, which is about 7.5% of the Mississippi River at maximum flow (*Restore The Mississippi River Delta, 2017*).

Sediment diversions are designed specifically to divert mineral sediments to the surrounding wetlands but will also divert a large volume of freshwater (*Restore The Mississippi River Delta, 2017*). Two existing freshwater diversions have already been built on the Mississippi River, emptying river water into Barataria and Breton basins. Davis Pond Diversion (DPD) was constructed in Barataria Bay in 2002 to prevent saltwater intrusion (Fig. 1). Caernarvon Freshwater Diversion (CFD) was opened in Breton Sound in 1991 to prevent encroaching high salinities from affecting commercial shellfish production (*Snedden et al., 2007*; *US Army Corps of Engineers, 2004*; Fig. 1). The CFD and DPD freshwater diversions are the primary restorative modifications that may be used to examine benthic community response to restored hydrologic connectivity along the Louisiana coastline. The comparison between existing freshwater diversions and proposed sediment diversions is modest at best because of the significant size difference between these restoration projects. The CFD can divert 226 $m^3 s^{-1}$ of Mississippi River water into Breton Sound, but the diversion's mean discharge is only 45 $m^3 s^{-1}$ and is pulsed to protect privately-owned land from incurring significant damage (*Restore The Mississippi River Delta, 2017*). DPD, which can divert only 300 $m^3 s^{-1}$, is about five times smaller in scale than the proposed sediment diversions (*Das et al., 2012*). Despite the difference in scale, we hope to gain insight from the impacts of these two existing diversions, as they may compare to the future impacts of the proposed sediment diversions. Impacts of the existing freshwater diversions on the benthic community are documented, and here they are compared to the potential impacts of proposed sediment diversions. Categories of impacts include (a) salinity modifications; (b) sedimentation; (c) nutrient loading and primary production, and; (d) implications for commercial species.

Benthic invertebrates are widely used as indicators of local environmental change because of their size, relatively long life spans, predictable response to stressors, and frequently sessile forms (*Montagna, Kalke & Ritter, 2002*; *Weisberg et al., 1997*; *Rosenberg & Pearson, 1978*). Many benthic fauna, such as brown shrimp and blue crabs, are also

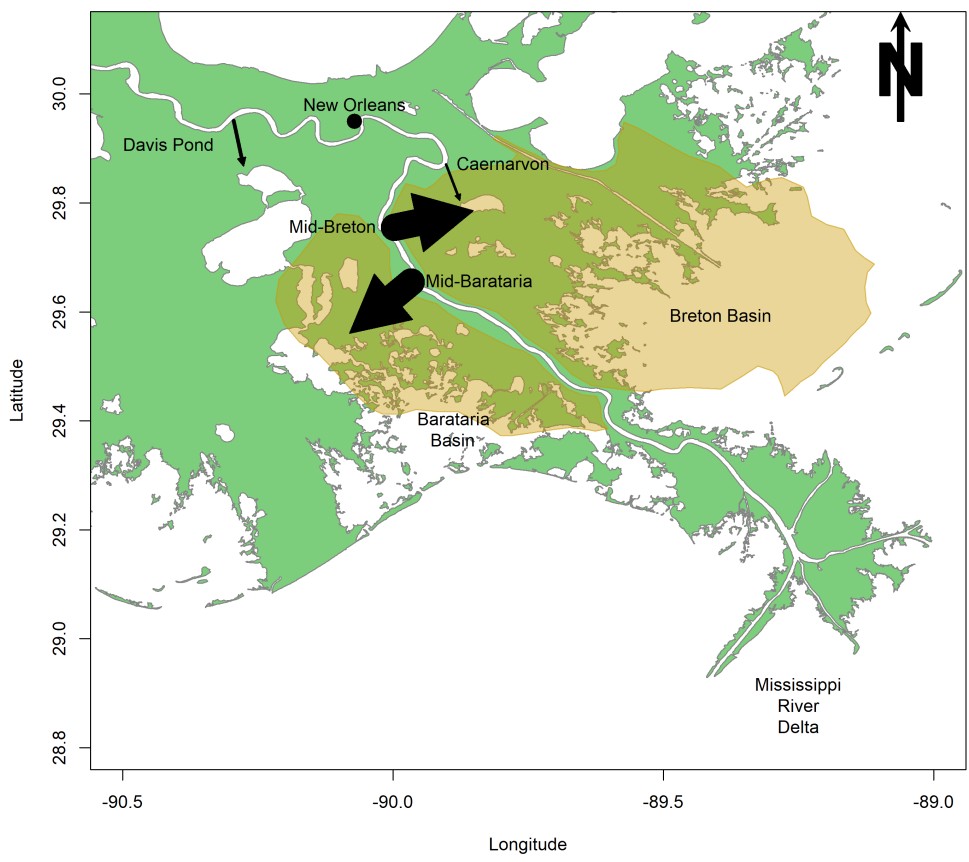

**Figure 1** **Map of the study area surrounding the lower Mississippi River.** Arrows indicate the magnitude and direction of flow for existing freshwater diversions (Caernarvon and Davis Pond) and proposed sediment diversions (mid-Barataria and mid-Breton). The highlighted region indicates the area of impact of the proposed sediment diversion.

of commercial value to Louisiana fisheries. However, there are few datasets available for benthic infauna in the northern GOM. *Whaley & Minello (2002)* examined the distribution of the benthic infauna community across the marsh surface, but their study was limited to one location in the polyhaline region of Galveston Bay, TX. None of the existing or planned long-term monitoring programs in Louisiana include sampling of coastal benthic infauna (*Hemmerling & Hijuelos, 2017*). There are some long-term studies of benthic infauna on the continental shelf (*Gaston & Edds, 1994*; *Harper & Rabalais, 1997*), but these are not relevant to estuarine systems. Benthos in coastal bays and estuaries throughout coastal Mississippi and Louisiana were sampled between 2001 and 2005 by the EPA, but this effort was discontinued and sampling effort was extremely limited in Barataria and Breton basins (*USEPA, 2007*). There have been no studies examining the impact of salinity reduction on benthic infauna communities and associated ecosystem functions and services in Louisiana's coastal basins. Of the benthic studies in the Gulf of Mexico, the vast majority are observational in nature, with only 5% of these studies including a manipulative or experimental component (*Brooks et al., 2006*). Along the Louisiana coastline, specifically in
estuaries and tidal bayous, benthic surveys and experiments remain an untapped research tool to assess environmental status.

## SURVEY METHODOLOGY AND OBJECTIVES

In light of the proposed sediment diversions, characteristics of the benthic community in coastal Louisiana as they pertain to areas of restored hydrologic connectivity are examined. This review consists of studies conducted in estuaries surrounding the Mississippi River Bird's foot Delta. Each study measured and/or modeled facets of benthic community response in an area reconnected to freshwater flow. Well-studied Texas estuaries are used here as reference ecosystems because they present a wealth of both survey and experimental information that spans several decades (Fig. 2). A traditional rather than a systematic review approach was taken because of the location-specific restoration goals that comprise the framework for this study and because benthic information within coastal Louisiana estuaries is sparse and uneven. Within the scope of this geographic area, the objectives of this review are to (1) examine the body of existing research on known drivers of benthic community response that are related to restored freshwater flow; (2) point out both achievements in the field and major knowledge gaps to be addressed before construction begins; and (3) provide recommendations for future experiments and surveys on coastal benthos that are designed to track ecosystem health and restoration success.

## SALINITY

### Salinity reduction

Salinity change is the most direct and measurable result of freshwater inflow into an estuary. At least some portion of the water contained in estuaries worldwide is brackish, and is subject to seasonal, spatial, and temporal patterns that result in natural salinity variation through press and pulse regimes (*Van Diggelen & Montagna, 2016*). Sediment diversions will be a significant source of freshwater to Barataria Bay and Breton Sound. Higher discharge, combined with small estuary residence times, will lead to reduced salinities in some areas, most predictably around the inflow, but also throughout other parts of the basin (*Allison & Meselhe, 2010*; Fig. 3). A coupled hydrology-hydrodynamics model of Barataria Bay after the opening of DPD indicated that the middle of the estuary was influenced most strongly by restored freshwater flow and experienced changes in average salinities of up to 10 ‰, whereas the upper and lower estuary remained relatively unaffected (*Das et al., 2012*). After the opening of CFD, average salinities just past the inflow area in Breton Sound were reduced by approximately 3 ‰ relative to reference site Bayou Terre Aux Boeufs (*Rozas et al., 2005*). These two models capture the current salinity regimes in these basins with the two existing freshwater diversions. With pending sediment diversion construction, it is likely that additional numerical models will be created to encompass the potential salinity scenarios in Breton Sound and Barataria Bay. These models should capture the difference in scale between proposed sediment diversions and existing freshwater diversions.

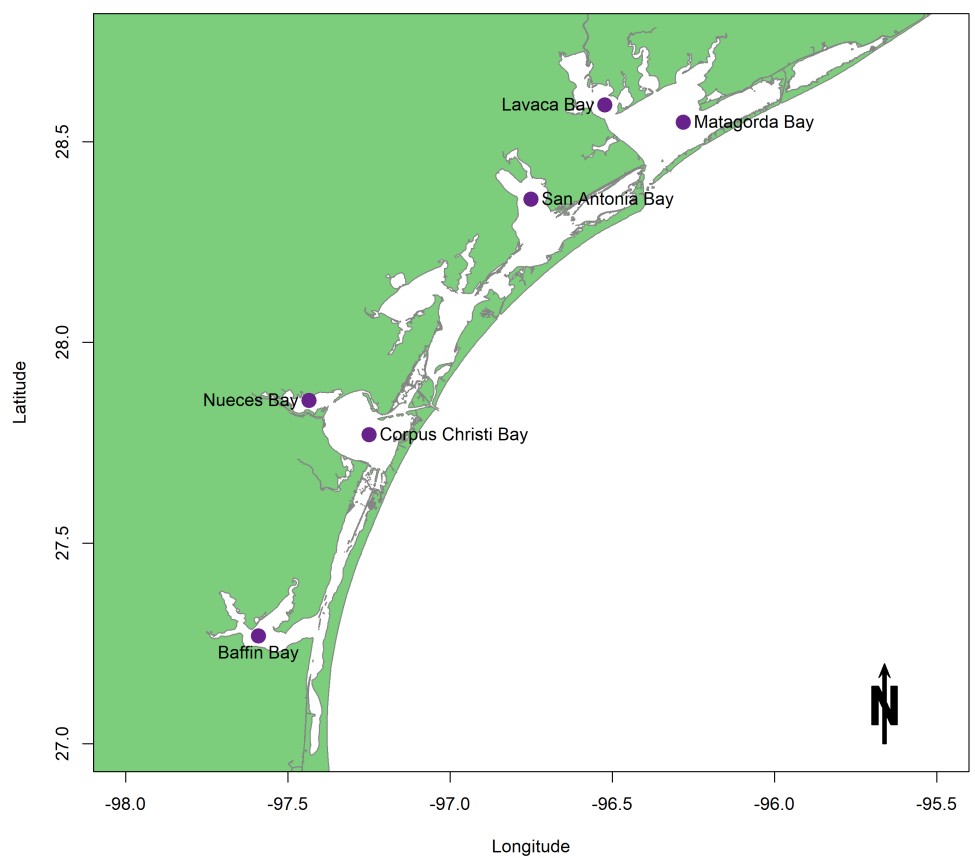

**Figure 2  Map of the Texas lagoonal estuary study area.**

## Benthic response to salinity

Estuarine salinity gradients impact the distribution of benthic invertebrates, as well as populations of commercially and recreationally important species that depend on invertebrates as a food source (Fig. 3). Sediment diversion operations will likely generate a steep salinity gradient, with oligohaline (fresh, 0.5–5.0 ppt) water directly around the inflow area (*Allison & Meselhe, 2010*). Estuaries generally support high biomass of benthic invertebrates, with varying salinity tolerances, but many are not tolerant to fresh conditions (*Fredette & Diaz, 1990*). Benthic infaunal macroinvertebrates (macrobenthos >one mm body length) may decline under oligohaline conditions due to high metabolic cost and mortality (*Duggan et al., 2014*). However, smaller opportunistic meiobenthos (benthic invertebrates between 0.063 mm and 0.5 mm body length) may increase under these conditions (*Alongi & Robertson, 1995*; *Giere, 2009*). Shifts in benthic macroinvertebrate distributions may inevitably cause contemporaneous shifts in their predators: blue crabs, shrimp, and demersal fish (*Laughlin, 1982*). If macroinvertebrates are not available in an area, food chains may shorten. Shrimp and other higher trophic level species may directly consume bacteria or detritus, rather than consuming intermediate trophic level organisms (macroinvertebrates) (*Alongi & Robertson, 1995*).

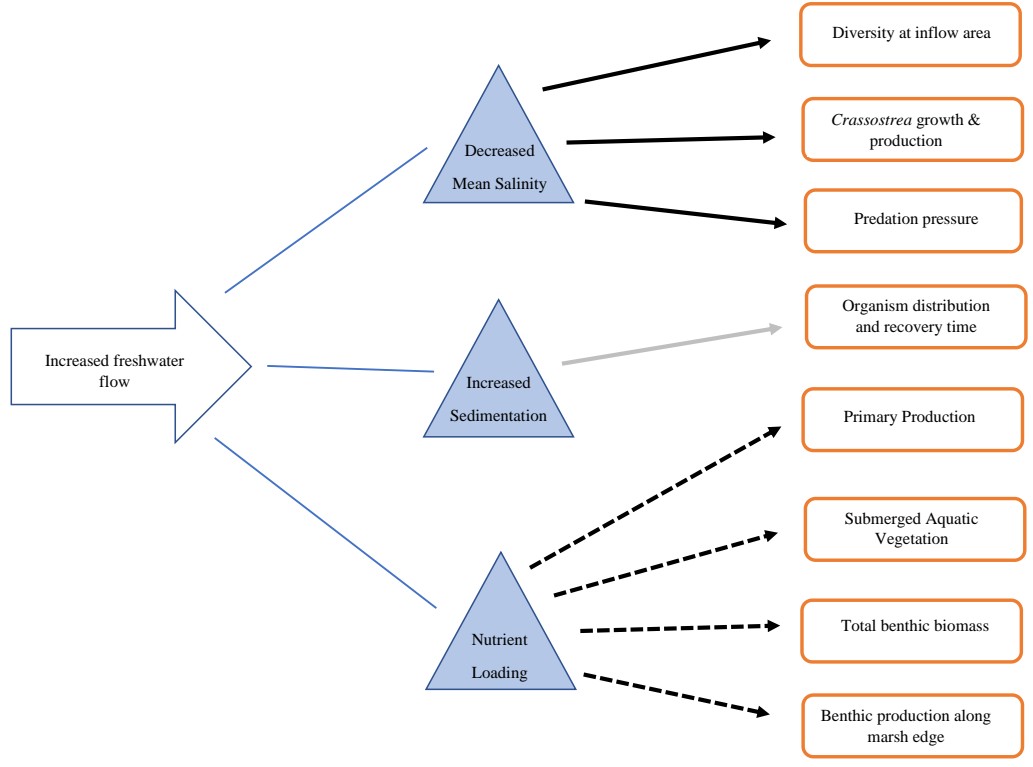

**Figure 3** **Conceptual diagram of the consequences of increased freshwater flow on the benthic community.** Solid black arrows indicate a decrease in the recipient, dashed arrows indicate an increase in the recipient, gray arrow indicates mixed/unknown effects in the recipient.

High flow conditions that reduce salinities may result in differences in both biomass and functional diversity of benthic invertebrates. Estuarine areas with initial high salinities that experience high discharge may see shifts in benthic invertebrates due to salinity change. Estuarine areas with initially low salinity that experience high discharge will not freshen, and benthic shifts that occur will not be due to salinity. In bays that rely on significant riverine input, suspension feeder biomass generally increases, and deposit feeder biomass decreases with freshwater flow (*Montagna, Kalke & Ritter, 2002*; *Kim & Montagna, 2012*; Fig. 2). Under the low flow conditions, the opposite trend is observed (*Kim & Montagna, 2012*; *Brooks et al., 2006*). The diversity of feeding guilds, such as suspension and deposit feeders, are often used to represent functional diversity of an ecosystem. Functional diversity decreased under high flow conditions in Texas estuaries through the loss of the deposit feeding guild (*Kim & Montagna, 2012*). In the context of the proposed sediment diversions, this situation may occur around the inflow areas characterized by low salinity and high nutrient levels; functional diversity in benthic invertebrates may be lost by a reduction in deposit feeders, but overall benthic biomass may increase or stay the same due to an increase in suspension feeders.

Low salinity can also have impacts on the metabolism of estuarine organisms, especially in systems heavily impacted by the stress of climate change. A major salinity threshold

between freshwater and estuarine organisms occurs around 5.0 ppt, because freshwater organisms generally cannot maintain correct osmotic pressure of their cells in salinity above 5.0 ppt, and marine organisms cannot maintain osmotic pressure in their cells in salinity below 5.0 ppt. (*Gray & Elliot, 2009*). Salinity change is one measurable stressor that may result from sediment diversion operation, but salinity change will often coincide with multiple stressors, such as changes in pH. Bivalves, which are calcifying organisms that are already impacted by lower pH in bodies of water worldwide, can regulate osmotic pressure in their cells at a cost (*Dickinson et al., 2013*; *Glaspie et al., 2018*). The metabolic cost of building tolerance to a stressor such as salinity is generally compounded when multiple stressors are present (*Glaspie et al., 2018*). The clam *Mercenaria mercenaria* as well as the oyster *Crassostrea virginica* have reduced ability to biomineralize their shells under both low salinity and acidification stress (*Dickinson et al., 2013*). This leaves them vulnerable to predation, and generally less fit for their environment (*Dickinson et al., 2013*; *Glaspie et al., 2018*).

Many benthic macroinvertebrates have reasonably wide salinity tolerances (*DeMutsert, 2010*), meaning that differences in average salinity generated by sediment diversions may not drive population shifts directly through mortality. The indirect effects of salinity change that will likely control macroinvertebrate distributions post-construction are reduced prey availability and increased metabolic cost. These controls will likely be disrupted by increasing freshwater to Barataria Bay and Breton Sound. Reduced prey availability, shortening of food chains (*Alongi & Robertson, 1995*), and higher metabolic cost will reduce macroinvertebrate biomass around the inflow area. Highly variable salinity and inundation is likely to result in low diversity characterized by dominance of opportunist species (*Palmer, Montagna & Kalke, 2002*). Even highly tolerant or mobile benthic invertebrates will have to contend with changes to the food web and metabolism as a result of diversion operation.

## Benthic recovery after high-flow events

Recommended operation for the planned sediment diversions include seasonal pulsed openings, which will generate seasonal high flow from the Mississippi River (*Peyronnin et al., 2017*). Pulse openings of CFD resulted in a steep salinity decline around the inflow area, and subsequent low diversity in the nekton community as compared to a reference site (*DeMutsert, 2010*). Previous studies of high flow events in Texas estuaries indicate that diversity will decrease if the salinity change is enough to disrupt the existing macrofaunal population, but abundance and biomass will likely stay the same (*Montagna, Kalke & Ritter, 2002*). Opportunists tend to quickly populate the vacated area, especially in areas where space limits carrying capacity (*Peterson, 1979*). These short-lived pioneers have high turnover rates and can be displaced when macroinvertebrate larvae recolonize the area, keeping abundance and biomass steady (*Peterson, 1979*). After a pulse from CFD, a 2–4 week lag time is needed for the salinity gradient to be restored to pre-pulse values (*Allison & Meselhe, 2010*; *Das et al., 2012*). However, the diversity of the original assemblage can take years to recover, because seasonal pulses may disrupt the area multiple times per year (*Borja et al., 2010*). Episodic high flow events that reduce salinities in certain areas may

not allow the full recovery of benthic assemblages near the inflow area of the sediment diversions.

## Riverine vs. marine influence

Estuarine benthic assemblages are often characterized by their proximity to a river or ocean. Salinity variability generally decreases from river to sea (*Kim & Montagna, 2012*; *Van Diggelen & Montagna, 2016*). Along the gradient from the upper estuary (closest to river outflow) to the lower estuary, salinity ranges become narrower, creating more stability and resulting in higher diversity, species richness, and evenness (*Kim & Montagna, 2012*; *Van Diggelen & Montagna, 2016*). The lower estuary also tends to have a closer connection to marine propagules, allowing the community to recover from disturbance more quickly than the upper estuary (*Kim & Montagna, 2012*).

Restored freshwater flow can result in a decrease in total benthic biomass in portions of the estuary with higher marine influence, and an increase in total benthic biomass in portions of the estuary with higher riverine influence. In several Texas lagoonal estuaries with no direct connection to the Gulf of Mexico, high freshwater input resulted in as much as a 70% increase in total benthic biomass (*Kim & Montagna, 2012*: Fig. 2). Both Breton Sound and Barataria Bay are larger and more open to the Gulf of Mexico than these well-studied Texas estuaries (*Kim & Montagna, 2012*; *Kim & Montagna, 2009*). There is evidence that the upper portions of Breton Sound and Barataria Bay rely more significantly on riverine inputs, and that the lower portions rely more significantly on marine inputs (*Allison & Meselhe, 2010*). It is possible that restored freshwater flow in Lousiana's deltaic estuaries will result in increased benthic biomass in the upper estuary, similar to Texas' lagoonal estuaries. However, the lower estuary may experience restored freshwater flow as a disturbance, resulting in declines in benthic diversity.

Studying the relationship between salinity and estuarine benthos can be approached by measuring average salinity or salinity variance, as both are drivers of benthic community response. Average salinity is more commonly used and is useful in a study area with a natural long-term salinity gradient, such as in Texas (*Montagna, Kalke & Ritter, 2002*: Fig. 2). These gradients can produce differences in endemic benthic assemblages. In one study, macrofaunal characteristics demonstrated a strong, approximately logarithmic relationship with average salinity (*Montagna, Kalke & Ritter, 2002*). Abundance peaked at salinity 32.7 ‰, biomass peaked at 18.7 ‰, and diversity peaked at 9.08 ‰ in a series of four Texas estuaries (*Montagna, Kalke & Ritter, 2002*; Table 1). Each measure of benthic community response has its own relationship to average salinity, but alone average salinity is not an adequate predictor of benthic response, because salinity over the long-term varies widely (*Kim & Montagna, 2012*).

Salinity variance may be a better predictor of community response than average salinity (*Van Diggelen & Montagna, 2016*). Salinity variance provides a means to understand the stability of an estuarine system. High salinity variance can influence diversity by creating unstable physical conditions for a complex community. Average salinity has positive and significant linear relationship with S (species richness) but possesses no relationship to H (Shannon-Weiner index) nor J (Pielou's Evenness index) in Texas estuaries (*Van*

**Table 1  Trends in nutrient loading, sedimentation, primary production and secondary production with distance from the mouth of a river.** Locations marked with an asterisk indicate that measurements were taken in at seasonally high flow times.

| Location | Measurement | Increasing distance from river mouth | Discharge ($m^3 s^{-1}$) | Reference |
|---|---|---|---|---|
| Breton Sound* | DIN | ↓ | 102 ± 49.1 | *Riekenberg, Bargu & Twilley (2015)* |
|  | DIN:P | ↓ |  |  |
|  | Chlorophyll α | ↑ (weak) |  |  |
|  | TP | = |  |  |
| Breton Sound | $\delta^{15}N$ (diversion) | ↓ | 50 | *Wissel & Fry (2005)* |
|  | $\delta^{15}N$ POM (diversion) | ↓ |  |  |
|  | $\delta^{15}N$ (grass shrimp) | ↓ |  |  |
| Rincon Bayou, TX | Benthic biomass | = | Post-diversion | *Montagna, Kalke & Ritter (2002)* |
|  | Benthic abundance | ↓ (slight) |  |  |
|  | Benthic diversity | ↑ |  |  |
| Lower Mississippi River | Sediment grain size between fine sand and medium sand | ↑ | ↑ | *Allison & Meselhe (2010)* |
| Breton Sound | Sediment deposition | ↓ |  | *Day et al. (2009)* |
|  | DIN | ↓ (57% over 20 km) |  |  |
|  | Chl α | Highest mid-estuary |  |  |
| Breton Sound | Phytoplankton biomass | ↓ | 45 | *DeMutsert (2010)* |
|  | SAV biomass | ↓ |  |  |
|  | Benthic algae biomass | ↓ |  |  |
|  | Nekton energy density | ↓ |  |  |
| Breton Sound | Stem density | ↑ | 184 | *Piazza & La Peyre (2007)* |
| Breton Sound | DIN:DIP | ↓ | 220 | *Lane et al. (2004)* |
|  | TN | ↓ (44%) |  |  |
|  | TP | ↓ (62%) |  |  |
|  | DIN | ↓ (57%) |  |  |
|  | DIP | ↓ (23%) |  |  |
| Barataria Bay | Ribbed mussel density | Highest in the middle |  | *Honig, Supan & La Peyre (2015)* |
|  | Ribbed mussel size | ↑ |  |  |
|  | Mussel recruitment | ↑ |  |  |
| Breton Sound | Sediment deposition | ↓ | 183 | *Wheelock (2003)* |
| Lavaca Bay, TX | Chl α | = |  | *Kim & Montagna (2012)* |
|  | Nutrient concentration | ↓ |  |  |

*Diggelen & Montagna, 2016*). In contrast, salinity variance has a strongly inverse relationship with each S, H, and J (*Van Diggelen & Montagna, 2016*). Species richness, diversity, and evenness have stronger relationships with salinity variance than with average salinity, although in general, estuaries with a higher average salinity also possess higher diversity (*Van Diggelen & Montagna, 2016*; *Gaston et al., 1998*). Estuaries with higher average salinity have lower salinity variance, and therefore more stability than areas with higher levels of intermittent freshwater flow (*Van Diggelen & Montagna, 2016*). After sediment diversions are constructed on the Mississippi, measuring both average salinity and salinity variance

as they relate to the benthic community will be useful. Relating these measurements to water residence time, size, and depth will produce a more complete picture of antecedent conditions that govern salinity structure. Characterizing salinity variance will require continuous monitoring. Closely monitoring these relationships with changing operations will furnish quality information that can be used for numerical models, as well as feedbacks for effective adaptive management for the large-scale restoration project.

## Salinity restrictions on marine predators

Increased freshwater flow can exclude marine transients and stenohaline species from preying upon benthic organisms (*Piazza & La Peyre, 2011*). Stenohaline species are physiologically restricted to a narrow salinity range (*Piazza & La Peyre, 2011*). High impact areas just outside CFD experience significant freshening and restrict marine species, but areas even just a few kilometers downstream of the diversion do not experience this phenomenon (*Piazza & La Peyre, 2011*). Several species of small forage fish experience changes in density and biomass in relation to freshwater pulses, whereas the abundance of their prey, including grass shrimp, increases in the absence of predators (*DeMutsert, 2010*; *Piazza & La Peyre, 2011*). In Barataria Bay, ribbed mussel densities are largest at mid-salinity marsh edge sites with high stem density of *Juncus roemerianus* (*Honig, Supan & La Peyre, 2015*; Table 1). High mid-salinity ribbed mussel densities may indicate a threshold for what their major marine predator, *Callinectes sapidus*, can tolerate (*Honig, Supan & La Peyre, 2015*). Salinity restrictions to marine transients may only prevent predators from using affected areas for weeks at a time (*Allison & Meselhe, 2010*; *Das et al., 2012*); thus, sessile benthic species in areas of seasonally variable salinity will not be able to rely on salinity restrictions for permanent protection.

Short-term endurance of stressful low salinities can have an energetic payoff for marine predators. Several studies have noted increases in prey species tolerant to lower salinities populating high impact areas near restored freshwater flow, creating an energetic subsidy for the area (*Piazza & La Peyre, 2012*). For example, many poeciliid species, specifically *Heterandria formosa* (least killifish) and *Gambusia affinis* (mosquitofish) increase in size, energy, and density after flood pulses through CFD (*Piazza & La Peyre, 2012*; *Piazza & La Peyre, 2007*). Marine predators occasionally capitalize on this concentration of prey. Juvenile and sub-adult red drum prefer mesohaline (15–20 ppt) conditions (*Dance & Rooker, 2016*). Anecdotal observations of red drum (*Sciaenops ocellatus*) foraging along the marsh edge and outside of their preferred salinity range (<1 ppt) after a flood pulse indicate that some predators will endure sub-optimal physical conditions to exploit a predictable (often seasonal) resource (*Piazza & La Peyre, 2012*). Though predators are scarce at high inflow areas, their rate of consumption may be higher, which could threaten sessile benthic organisms that are unable to avoid predators. Understanding the impacts of proposed sediment diversions on benthic communities requires an examination of the role of predators in these systems. Defining the relationship between salinity endurance and foraging behavior of predators on the benthic community is an area for further research.

## SEDIMENTATION

### Long-term sediment deposition from pulsed freshwater flow

Sediment delivery to the marsh surface is dependent on the suspended sediment load, as well as other conditions such as prevailing winds, water level, and velocity (*Day et al., 2009*). In Breton Sound, pulsed flow regimes resulted in increased sediment deposition rates compared to press flow regimes at distances less than 6 km from CFD (*Wheelock, 2003*; *Day et al., 2009*). Short-term deposition rates as high as 4.740 g m$^{-2}$ d$^{-1}$ were measured from pulsed flow water velocity >183 m$^3$ s$^{-1}$ (*Wheelock, 2003*). Mineral sediments were deposited closest to the inflow area, and a higher percentage of organic matter was deposited in sediments further downstream (*Wheelock, 2003*). In marshes within 6 km of CFD, sediment delivery is determined primarily by the fluvial pulses, whereas the deposition onto marshes outside of this range is governed both by fluvial pulses and sediment resuspension (*Wheelock, 2003*). The annual rate of sedimentation from CFD is generally low, with some estimates suggesting it is currently keeping pace with RSLR (*Day et al., 2009*), and others suggesting that this rate of land-building is 66% deficient to keep pace with the combined effects of sea level rise, erosion, and subsidence (*Wheelock, 2003*). Currently, no studies examine how these relatively slow deposition rates affect the benthic community in this area. Fourleague Bay, TX which can be used as a proxy for natural rates of land-building, has a very high rate of sediment particulate deposition (0.11–0.18 cm/day) and a lower long-term rate of accumulation (0.59–0.39 cm/yr) and benthic oxygen uptake (a proxy for benthic productivity) is independent of both of these rates (*Teague, 1983*). The planned sediment diversions will exceed the current sediment deposition in Breton Sound at a rate unknown, but the area of greatest concern to the benthic community will be the area immediately adjacent to the sediment diversion outlet due to high deposition rates of mineral sediments in these areas.

Planned sediment diversions are designed to specifically deliver mineral sediments to the surrounding marsh. The largest median sediment grain size is expected closest to the inflow area, and smaller grain size sediments are expected downstream in lower flow areas. Grain size distribution is a determinant of benthic community distribution and is subject to change with new flow regimes. Large grain size sediments generally support high drainage, circulation, and oxygen content within the soils (*Gray & Elliot, 2009*). Smaller grain size sediments have lower permeability, lower oxygen, and more organic matter than large grain size soils (*Gray & Elliot, 2009*). These factors, among others controlled by sediment properties, can influence the distribution of benthic organisms (*Sanders, 1958*). Restored flow in Barataria and Breton basins may increase habitat quality for macrofauna in the upper reaches of the estuary, close to the sediment diversion outlet; however, it remains to be seen how sediment grain size, deposition, salinity, and a number of environmental variables will interact to ultimately drive benthic biomass and diversity in the affected estuaries.

### Benthic recovery time from major sedimentation events

Recommendations on Mississippi sediment diversion operations from a technical working group include a gradual increase of flow for the first 5–10 years of operation to minimize
disturbance to wildlife (*Peyronnin et al., 2017*). Extreme pulses of sediments are to be avoided; rather, a distributary network should be gradually formed from pulse openings throughout the winter and early spring (*Peyronnin et al., 2017*). Once this distributary network is established, erosion of sediments will likely be reduced, and smothering of benthic organisms will be less probable. Until that point, every pulse opening could serve as a disturbance event of extreme sediment erosion and deposition that impacts the benthic community. Thus, ecosystem managers interested in predicting short-term changes in the benthic community may be able to use previous studies on dredging or storms.

Bed movement, deposition, and erosion of sediments can strongly affect benthic organisms (*Miller, Muir & Hauser, 2002*). Most benthic organisms reside within the top 10 cm of the sediment, and can tolerate slow rates of sediment delivery or removal (*Miller, Muir & Hauser, 2002*). Dredging and events of extreme sediment deposition can smother organisms that are sensitive to burial, allowing pioneer species like small, mobile polychaetes to reestablish the area (*Borja et al., 2010*; *Miller, Muir & Hauser, 2002*). Thus, severe sedimentation or erosion events are more likely to result in alterations in community structure rather than loss of benthic biomass according to studies conducted in Delaware Bay (*Miller, Muir & Hauser, 2002*). Recovery of the benthic community is a function of the rate of sediment deposition, and the recruitment of new organisms to the area (*Borja et al., 2010*). Meiofauna can recover from a sediment disposal event in as little as 3–18 months, but macrofauna typically take 2–5 years to recover (*Borja et al., 2010*). Areas disturbed by restored freshwater flow in Louisiana will likely be recolonized quickly by opportunistic species, but the time needed for a complete recovery and the composition of the resulting community cannot be predicted without an understanding of the current community composition, benthic settlement rates, and composition of new recruits.

Storm systems alter sediment movement through increased erosion, resuspension, and deposition, and therefore may be comparable to planned sediment diversions. Benthic macroinfauna were negatively impacted by Hurricane Katrina, but salinity change, rather than sediment alteration, was likely responsible (*Engle, Hyland & Cooksey, 2009*). Another study observed lower abundances of surface-dwelling organisms following a significant storm event, but noted that sediment disturbance from severe, episodic storms do not affect benthic community structure any more than year-to-year variability (*Posey et al., 1996*). A third study observed significant decreases in diversity and abundance of benthic organisms after winter storms, and higher densities of opportunistic polychaetes in the several months following, indicating a major disturbance to the benthic community (*Hernández-Arana et al., 2003*). Deep burrowers have been known to migrate vertically and re-emerge after a dramatic deposition event (*Miller, Muir & Hauser, 2002*; *Posey et al., 1996*). In general, organisms that do not have a large degree of mobility, such as sessile surface-dwelling organisms or slow-burrowing infauna, are most impacted by sediment deposition because they cannot migrate vertically to avoid being smothered (*Posey et al., 1996*). Two Chesapeake Bay studies showed that the population of *Mya arenaria*, a deep-burrowing bivalve with relatively little mobility as an adult, was negatively impacted by Tropical Storm Agnes, resulting in mass mortality for the species (*Glaspie et al., 2018*). Evidence from storms indicates that sediment deposition may have severe and lasting

consequences for the benthic community and sensitive organisms, depending on the rate of deposition and physiological constraints of the different species in the benthic assemblage (*Glaspie et al., 2018*).

## NUTRIENT LOADING AND PRIMARY PRODUCTION

### Trends with increased freshwater flow

The Mississippi River drains 41% of the continental United States, and it can be reasonably predicted that any diversion from it will contain significant nutrients from runoff (Fig. 3). Nitrogen and phosphorus from this runoff largely control production in coastal areas (*Lane et al., 2004*). Nitrogen is positively correlated with freshwater discharge (*Lane et al., 2004*; *Riekenberg, Bargu & Twilley, 2015*). Both riverine discharge and nitrogen input are highest during the winter and lowest during summer in Breton Sound (*Riekenberg, Bargu & Twilley, 2015*; Table 1). Phosphorus is a limiting nutrient and there is no seasonal variation in phosphorus input to coastal systems (*Riekenberg, Bargu & Twilley, 2015*). When estuary turnover is high, chlorophyll $\alpha$ can have a roughly inverse relationship with discharge, which expels phytoplankton from the estuary as they are generated (*Riekenberg, Bargu & Twilley, 2015*). In the case of Breton Sound, nearly half of the annual water turnover in the upper estuary occurs during the 4 weeks of pulsed flow from CFD (*Day et al., 2009*). During low flow times, rates of phytoplankton production exceed rate of turnover, and the highest densities of phytoplankton are found mid-estuary (*Riekenberg, Bargu & Twilley, 2015*; *Day et al., 2009*). Water flow is an important control on primary production in Breton Sound when nutrients are not limiting. In estuaries with tighter nutrient cycling, chlorophyll $\alpha$, and nutrients decrease in proportion to decreasing freshwater inputs and vice versa (*Kim & Montagna, 2012*; *Day, Madden & Twilley, 1994*; *Mortazavi et al., 2012*). Breton Sound and Barataria Bay do not exhibit tight nutrient cycling, and when sediment diversions are operable, nutrients will likely be in excess (*Day, Madden & Twilley, 1994*; *Day et al., 2009*). Optimized operations will emit pulses of relatively high nutrient water. Nutrient loads transported into Breton Sound and Barataria Bay will be high, but primary production may still be controlled by water flowing out of each basin. At times when both flow and primary production are high, the most resources will likely be available for filter feeders.

### Secondary production results from nutrient loading

Enhanced secondary production, defined as the biomass resulting from consumers in an ecosystem, may be a measurable effect of nutrient loading (Fig. 3). Nutrient loading predictably increases abundance and biomass of benthic infauna. (*Montagna & Yoon, 1991*). At the upper, river-dominated end of San Antonio Bay, TX, benthic primary production was five times higher than at the lower end (*Montagna & Yoon, 1991*: Fig. 2). Grazing rates were also 2–5 times higher at the upper end (*Montagna & Yoon, 1991*). Grazing rates on microalgae were 2–6 times higher than primary production, indicating that riverine nutrient inputs stimulate populations of grazers (*Montagna & Yoon, 1991*). Benthic grazing rates can be used to understand how nutrient inputs stimulate both primary and secondary production (*Montagna & Yoon, 1991*). However, baseline grazing rates have

not been measured in Barataria and Breton basins. This information will allow coastal researchers to understand the roles of the benthic and pelagic food webs and monitor how trophic transfer changes with restored river flow.

High nitrogen inputs from the Mississippi River will be rapidly taken up by the productive estuarine ecosystem before reaching the Gulf of Mexico (Fig. 3). After a flood pulse, dissolved inorganic nitrogen (DIN) decreases approximately 5-fold from the opening of CFD to the Gulf of Mexico (*Lane et al., 2004*; Table 1). Uptake into marsh food webs, burial, denitrification, and dilution are responsible for this observed diminution (*Wissel & Fry, 2005*; *Riekenberg, Bargu & Twilley, 2015*). The attenuation of DIN after an event of high riverine input is confirmed by a host of studies (*Wissel & Fry, 2005*; *Riekenberg, Bargu & Twilley, 2015*; *Perez et al., 2011*; *Teague, 1983*). Riverine inputs from CFD are credited for 75% of the food web support in upper Breton Sound, and 25% mid-estuary (*Wissel & Fry, 2005*). In a stable isotope analysis mixing model, grass shrimp near the inflow area of CFD were enriched in $\delta^{15}$N which is associated with anthropogenic nutrient inputs from the Mississippi River (*Wissel & Fry, 2005*; Table 1). Along the same sampling transect, an enrichment in $\delta^{13}$C indicated an increase in primary production, which was reflected in grass shrimp biomass, as they mainly consume epiphytic algae (*Wissel & Fry, 2005*). Since not all benthic fauna are fully dependent on epiphytic algae for dietary support, this model is likely an overestimate of how nitrogen enrichment contributes to the benthic food web, but it provides an informative baseline for further studies. Trophic transfer studies incorporating more benthic species, particularly those of varying trophic levels, will be necessary to monitor the lengths of the food chains with the pending changes in nutrient availability.

## Primary production along the marsh

Higher rates of primary production were measured for marshes surrounding the Mississippi River delta (1850 g C/m$^2$ per year) compared to open water (445 g C/m$^2$ per year), confirming that the areas which will receive input from sediment diversions are highly productive (*Bahr, Day & Stone, 1982*; Table 1). During times of high freshwater inflow, primary production mainly occurs at the fresh-saline interface, along bay margins and in tidal bayous (*Day, Madden & Twilley, 1994*). Freshwater pulses into Breton Sound distribute basal resources, which are exploited by consumers that quickly assemble in these newly accessible areas (*Piazza & La Peyre, 2012*).

Timing sediment diversion operation to take advantage of winter cold fronts will enable sediment resuspension and deposition on the marsh surface, and reduce impacts on dormant vegetation (*Peyronnin et al., 2017*). Nutrient-rich waters will inundate the marsh surface, expediting nitrogen removal and burial, although denitrification potential is low during the winter months (*Peyronnin et al., 2017*). Diatom enrichment, specifically an increased ratio of centric to pennate diatoms, may also be observed in areas of low salinity and high sedimentation (*Falcini et al., 2012*). If estuary flushing rates are high, excess nitrogen and phytoplankton will flow into the Gulf of Mexico. Primary and secondary production will have the potential to increase along bay margins and in tidal bayous (*Day, Madden & Twilley, 1994*). If estuary flushing rates are low, excess nitrogen will be

taken up during the growing season by soil microbes, macrophytes, marsh vegetation and other primary producers (*Peyronnin et al., 2017*). Low flushing rates paired with high nutrient freshwater input could cause an estuary-wide water level rise, which has negative consequences. Water level rise will inundate the marsh surface, releasing soluble reactive phosphorus, and creating suitable conditions for cyanobacteria responsible for harmful algal blooms (HABs) (*Zhang, White & DeLaune, 2012*; *Riekenberg, Bargu & Twilley, 2015*). These conditions are likely to negatively impact benthic production. Estuary flushing rates should be monitored closely to predict these conditions as they arise.

## Habitat substrate increases local secondary production

Freshwater inflow and increased nutrient inputs from the Mississippi River have increased the presence of submerged aquatic vegetation (SAV) in the form of rooted vascular plants and macroalgae (*Rozas et al., 2005*). Mean winter salinity, turbidity, and exposure are the main drivers for the presence of SAV (*DeMarco et al., 2018*). Following freshwater releases from CFD, a 66% increase in SAV was observed at inflow sites, which experienced an average salinity decrease from 8.3 ‰ to 5.9 ‰, and an 18% increase was observed at reference site Bayou Terre Aux Boeufs (*Rozas et al., 2005*). *Myriophyllum spicata* and *Potamogeton pusillus* were the most abundant SAV species, but *Ruppia maritima*, *Ceratophyllum demersum*, and *Najas guadalupensis* were also common (*Rozas et al., 2005*). In response, crustacean densities were 1.6 and 3.0 times higher, respectively, compared to non-vegetated bottom sites (*Rozas et al., 2005*). A similar trend in non-seagrass SAV was observed in a newly restored tidal marsh in the Sacramento-San Joaquin Delta, but the restored marsh SAV was vulnerable to invasive species of fish (*Grimaldo et al., 2012*). The marsh edge provides a similar, structurally complex habitat in this estuarine environment for benthic fauna. Benthic larvae also experience increased settlement out of the water column as flow slows over seagrass beds and marsh edge, abetting recruitment in these areas (*Chaplin & Valentine, 2009*). Juvenile blue crab recruitment in hard bottom substrates is nearly zero due to predation, while recruitment into complex substrate is slightly higher (*Heck, Coen & Morgan, 2001*). Preference for marsh edge vs. SAV varies by species, but all species preferred vegetated areas over non-vegetated areas (*Castellanos & Rozas, 2001*; *Rozas et al., 2005*). Overall, in studies conducted in northern Gulf of Mexico estuaries, the presence of SAV led to changes in macrofaunal abundance and density, but not in community composition (*Rozas et al., 2005*). The presence of SAV has been strongly connected to secondary production in this geographic region because it serves as complex habitat, providing protection and providing niches for a wide variety of species (*Chaplin & Valentine, 2009*). An SAV Likelihood of Occurrence (SLOO) model was developed, which has a 74% accuracy in predicting the presence of SAV along the Louisiana coastline (*DeMarco et al., 2018*). This model will likely be useful for further benthic studies before and after the sediment diversions are constructed. Other potential benefits of SAV to the benthic community, such as increased water column dissolved oxygen and facilitation of larval recruitment have not been studied in depth in this region (*Rozas et al., 2005*; *Chaplin & Valentine, 2009*; *Heck, Coen & Morgan, 2001*).

## Habitat substrate decreases abundance of large predators but increases abundance of small predators

For the first 5–10 years of proposed pulsed operation, flowing water will channelize the marsh, augmenting marsh edge habitat (*Peyronnin et al., 2017*). Opening of sediment diversions is projected to cause ephemeral inundation along the marsh edge, creating a shallow habitat that physically restricts large predators (*Piazza & La Peyre, 2007*). Juvenile brown shrimp, white shrimp, blue crab, and red drum use this habitat as a nursery because of the protection from predators it provides, as well as for access to organic matter, small fish, and benthic invertebrates. Marsh edge habitat has a lower density of predators, and contains high proportions and abundances of smaller individuals, and high species richness (*La Peyre & Birdsong, 2008*; *DeMutsert, 2010*). Red drum, black drum, and blue crab are well-documented predators of benthic infauna in coastal Louisiana (*Scharf & Schlicht, 2000*; *Kim & Montagna, 2012*); however, adults of these species are seldom observed in these shallow areas with short flooding duration, likely due to water depth restrictions (*Piazza & La Peyre, 2007*). Large predators prefer a deep and complex habitat to forage in as well as hide, specifically one with morphological irregularity (*La Peyre & Birdsong, 2008*). The absence of large predators and the rapid assimilation of incoming nutrients is evidence for strong bottom-up trophic controls along the marsh edge in estuaries such as Barataria and Breton (*Piazza & La Peyre, 2012*; *Wissel & Fry, 2005*). This phenomenon is projected to augment the survivorship of oysters, small forage fish, and juvenile species in this area (*DeMutsert, 2010*; Table 2). Several predictive models exist for predator diets in Barataria and Breton basins, but there are few observational or experimental studies to validate predictions (*DeMutsert, 2010*). Trophic transfer data including stomach contents and stable isotope analyses of these predatory species will be needed to monitor their feeding habits in emerging marsh edge habitats.

In these same shallow areas of marsh inundation, large predators are restricted, but small predators are present in high abundance (*La Peyre & Birdsong, 2008*; *DeMutsert, 2010*). Small, omnivorous fish, such as the sheepshead minnow and striped mullet, rapidly colonize the flooded marsh surface feed and on detrital matter, epiphytic bacteria, insect larvae, and benthic infauna as they seek refuge from large predators (*Piazza & La Peyre, 2007*; Table 2). Juvenile red drum and blue crabs also utilize this nursery habitat (*Scharf & Schlicht, 2000*; *Beseres & Feller, 2007*). These inhabitants can both prey upon and compete for resources with benthic fauna (*La Peyre & Birdsong, 2008*; *DeMutsert, 2010*; *De Mutsert, Cowan & Walters, 2012*). Benthic invertebrates and small, demersal fish are chronically understudied in this region, leaving large gaps in our knowledge of their trophic interactions. Actual trophic transfer data is needed to define their relationship with one another as well as with large predators (*Piazza & La Peyre, 2007*; *Piazza & La Peyre, 2012*). Incorporating these species into existing numerical models will further elucidate the estuarine food web and inform the Adaptive Management Plan for sediment diversion operation.

**Table 2  Impacts of river discharge on characteristics of benthic predator and prey species.** The direction of impact on each species/taxa is relative to a reference area that is relatively unaffected by river discharge.

| Reference | Species/Taxa | Discharge | Salinity (‰) | Measurement | Direction of Impact |
|---|---|---|---|---|---|
| *De Mutsert, Cowan & Walters (2012)* | Penaeid shrimp (benthic prey) | High | 1.4 ± 3.6 ‰ | Biomass | ↑ |
| | | Medium | 6.0 ± 4.6 ‰ | | = |
| | | Low | 10.3 ± 3.6 ‰ | | = |
| *DeMutsert (2010)* | Black drum (predator) | High | 1.4 ± 3.6 ‰ | Biomass | ↑ |
| | | Medium | 6.0 ± 4.6 ‰ | | ↓ / = |
| | | Low | 10.3 ± 3.6 ‰ | | ↓ |
| *DeMutsert (2010)* | Red drum (predator) | High | 1.4 ± 3.6 ‰ | Biomass | ↓ |
| | | Medium | 6.0 ± 4.6 ‰ | | ↓ |
| | | Low | 10.3 ± 3.6 ‰ | | ↓ |
| *DeMutsert (2010)* | Blue crab | High | 1.4 ± 3.6 ‰ | Biomass | ↑ |
| | | Medium | 6.0 ± 4.6 ‰ | | ↓ |
| | | Low | 10.3 ± 3.6 ‰ | | ↓ |
| *DeMutsert (2010)* | Sheepshead minnow | High | 1.4 ± 3.6 ‰ | Biomass | ↓ |
| | | Medium | 6.0 ± 4.6 ‰ | | = |
| | | Low | 10.3 ± 3.6 ‰ | | = |
| *DeMutsert (2010)* | Zoobenthos | High | 1.4 ± 3.6 ‰ | Biomass | ↑ |
| | | Medium | 6.0 ± 4.6 ‰ | | ↓ |
| | | Low | 10.3 ± 3.6 ‰ | | ↓ |
| *Piazza & La Peyre (2011)* | Killifish | High impact | Both <5 | Abundance | ↓= |
| | | Low impact | | | |
| *Piazza & La Peyre (2011)* | Palaemonetes sp. (grass shrimp) | Restored | Both <5 | Abundance | ↑ |
| *Piazza & La Peyre (2011)* | L. setiferus (white shrimp) | Restored | Both <5 | Abundance | ↓ |
| *Piazza & La Peyre (2007)* | Riverine grass shrimp (benthic prey) | Inflow area | 0.3 ± 0.01 | Density | ↑ |
| *Rozas et al. (2005)* | Blue crab | Inflow area | 5.9–7.8 ‰ | Density | ↑ |
| *Rozas et al. (2005)* | Riverine grass shrimp | Inflow area | 5.9–7.8 ‰ | Density | ↑ |
| *Rozas et al. (2005)* | Palaemonetes spp. | Inflow area | 5.9–7.8 ‰ | Density | ↓ |
| *Rozas et al. (2005)* | Brown shrimp | Inflow area | 5.9–7.8 ‰ | Density | = |
| *Rozas et al. (2005)* | All crustaceans (10 species) | Inflow area | 5.9–7.8 ‰ | Density | ↓ |

# IMPLICATIONS FOR COMMERCIAL SPECIES

## Salinity and Crassostrea virginica

The relationship between the benthic community and salinity is a complex one, and not all species respond to salinity variability in the same manner. Biomass for some ecologically and economically important species increase in proportion to increased freshwater flow primarily because of lower predation pressure and increased habitat access for smaller individuals (*DeMutsert, 2010*). After the opening of CFD in 1991, species biomass distributions for *Micropterus salmoides* (largemouth bass), *Micropogonias undulatus* (Atlantic croaker), *Brevoortia patronus* (Gulf menhaden), *Farfantepenaeus aztecus* (brown shrimp) and *Litopenaeus setiferus* (white shrimp) all exhibited this increase (*DeMutsert, 2010*; *Day et al., 2009*). In contrast, *Crassostrea virginica* may be negatively

impacted by restored flow if pulsed flow drops the salinity below a tolerable range for oysters for an extended period (*Wang et al., 2017*; *DeMutsert, 2010*; *Day et al., 2009*). Both Breton Sound and Galveston Bay, TX experience average salinities on the lower range of oyster tolerance (11.7 ‰, and 12–19 ‰, respectively) (*Wang et al., 2017*; *Turner, 2009*). Oyster populations in estuaries with higher average salinities may be buffered from detrimental freshening, but in estuaries with a low average salinity, they are vulnerable, explaining why there is no conflict between the findings of *Turner (2009)* and *Buzan et al. (2009)* (*Turner, 2009*; *Buzan et al., 2009*; *Wang et al., 2017*). In another study, which modeled the response of the oyster population in Breton Sound to either potential sediment diversions or to RSLR, both oyster growth and production declined in almost any restored flow scenario (*Wang et al., 2017*). Oyster populations were most negatively impacted by large-scale sediment diversion (growth rate reduction of 125%; production decrease of 60%), then by high RSLR (growth rate reduction of 80%, production decrease of 36%), and moderately negatively impacted by small-scale diversion (growth rate reduction of 19%, production decrease of 5%). (*Wang et al., 2017*). This projected decrease is not an effect of salinity change alone. The above model incorporates salinity as well as circulation, water quality, temperature, bottom type, disease, and predation into this projection, but also notes that salinity and temperature are key drivers of oyster metabolism (*Wang et al., 2017*).

## Salinity and Recruitment on *Farfantepenaeus aztecus*

*Farfantepenaeus aztecus* (brown shrimp), another commercial species in this region, have a mixed relationship with increased freshwater flow in Breton Sound, where freshwater pulses can cause salinities to drop below 5 ‰ (*Wang et al., 2017*). A stable isotope analysis study showed that brown shrimp were displaced into higher salinity areas by increased freshwater flow from CFD but showed no significant changes in abundance or individual size (*Rozas et al., 2005*). Salinity may also have an indirect effect on brown shrimp growth by altering prey availability (*Rozas & Minello, 2011*). At peak flow during April and May, after the sediment diversions are built, there may be few high salinity habitats left for brown shrimp (*Das et al., 2012*). A model for penaeid shrimp, notes a slightly positive relationship between lower salinities in Breton Sound and brown shrimp biomass which may be due to a reduction in predation pressure (*De Mutsert, Cowan & Walters, 2012*). Brown shrimp populations are governed by salinity both directly (through recruitment) and indirectly (through prey and habitat availability). The success of the brown shrimp population in Breton Sound and Barataria Bay will depend on these combined effects of salinity, which have not been evaluated in depth.

Since *Farfantenenaeus aztecus*, along with many others, spawn during the spring, an estuary-wide salinity decrease along with prevailing climate conditions may jeopardize recruitment by pushing individuals further out toward the barrier islands, where habitat is less desirable (*Adamack et al., 2012*). Recruitment of benthic species in this geographic region is the least studied of the drivers of benthic response to freshwater flow. Recruitment of larval benthic organisms is the primary mechanism that allows benthic fauna to recover from disturbance and is also responsible for maintaining stocks of local populations

in Barataria Bay and Breton Sound (*Palmer, 1988*; *Flemer & Champ, 2006*). Many species recruit into these basins during some portion of their life cycle. Barataria Bay alone accounts for 44% of the inshore brown shrimp harvest in Louisiana (*Peyronnin et al., 2017*). Brown shrimp spawn offshore and then recruit into both Barataria and Breton basins in the early spring as north-flowing currents carry post-larvae from offshore into the estuaries (*Rogers et al., 1993*; *Saoud & Davis, 2003*). Juveniles then reside in these estuaries from approximately February to July (*Peyronnin et al., 2017*). Strong river discharge and cold fronts may interrupt this recruitment by pushing against the prevailing current carrying larvae (*Leo et al., 2016*). A delay in brown shrimp recruitment as much as one month may be observed (*Leo et al., 2016*). Recommended operation of sediment diversions limits the strongest river discharge to the winter, so that interference with brown shrimp larval recruitment is minimized (*Peyronnin et al., 2017*). Brown shrimp recruitment is relatively well understood due to their ecological and economic value to Louisiana.

## CONCLUDING REMARKS

Benthic community response to the impacts of sediment diversions is likely to be significant, based on benthic community response in geographically similar areas with restored freshwater flow. Salinity variance is an important predictor of benthic community response, as is average salinity. The area that will experience the greatest salinity variance is mid-estuary. Areas of the estuary which rely primarily on riverine input and are relatively removed from marine influence will likely experience an increase in benthic biomass as a result of restored freshwater flow, but the immediate inflow areas of new diversions will likely experience temporary or sustained low diversity, depending on diversion operation and the potential for recovery. The presence of SAV as a habitat for benthic organisms is positively impacted by increase freshwater flow and will affect recruitment and survivorship of benthic organisms. Increased nutrient input from the Mississippi River will be taken up along productive tidal bayous and bay margins, which will lead to an increase in both primary and secondary productivity in these areas. Sediment deposition from sediment diversions has perhaps the strongest potential to negatively impact the benthic community through smothering and burying, but even high rates of sediment deposition may be tolerable to the benthic community. Lastly, predation pressure is predicted to be lower due to salinity restrictions on predators and reduced physical accessibility or new marsh habitat.

## FUTURE RESEARCH RECOMMENDATIONS

(1) Studies on the effects of restored freshwater flow on the benthic community in estuaries along the Gulf of Mexico are geographically uneven. The aforementioned group of Texas estuaries has over two decades of consistent benthic surveys and experimental research, and are better characterized than Breton Sound, Barataria Bay, and other estuaries surrounding the Mississippi River. Regular benthic sampling before and after the construction of the proposed diversions is needed to assess the restoration effects on the benthic community.

(2) The response of relatively large demersal fish and crustaceans to increased freshwater flow has been surveyed, modeled, and analyzed in many quantitative and qualitative capacities. In many of these studies, a single species is chosen to represent the benthic food web, and is often of commercial value to Louisiana. Although these studies provide a baseline for further work, they are likely inadequate to understand the benthic food web.

(3) A dearth of trophic transfer and predator foraging behavior data in Breton Sound and Barataria Bay has led to a cloudy relationship between the direct effects of increased freshwater flow and the synchronous consequences in the estuarine food web, specifically the benthic food web. Due to the projected reduced predation pressure on small forage fish in Breton Sound and their anticipated increase in density, a better understanding of the interactions between these fish and benthic organisms would be particularly valuable.

(4) Mid-estuary salinity decreases, high estuary turnover during spring months, and the presence of SAV will likely impact larval recruitment of benthic species, but few studies examine the combined effects of these projected changes. Many benthic organisms spawn in the spring, when sediment diversion output will be highest, so it is critical that we understand recruitment mechanisms in this area so that we may project how they will be affected by diversions.

### Funding
The authors received no funding for this work.

### Competing Interests
The authors declare there are no competing interests.

### Author Contributions
- Jillian C. Tupitza and Cassandra N. Glaspie analyzed the data, prepared figures and/or tables, authored or reviewed drafts of the paper, and approved the final draft.

### Data Availability
    This is a literature review article and did not generate raw data.

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
