# Peer review of "Restored freshwater flow and estuarine benthic communities in the northern Gulf of Mexico: research trends and future needs"

_PeerJ, doi:10.7717/peerj.8587_

## Round 0.1 · original submission · Minor Revisions

I concur with both reviewers that this manuscript requires only minor revisions and that this review conveys a novel perspective that deserves greater attention.

Reviewer 1 ·

Basic reporting

There appears to be proposals to divert river flow to deliver sediment downstream to rebuild wetland habitats, and this study provides a review of the potential effects on benthic communities. This is well justified but should mention the many studies that describe wetland loss rates that have already happened. As written, the first paragraph makes it appear that it is a relative SLR issue alone, but I thought it was due to scouring effects of the dykes and levees along the Mississippi River and this has been going on for a very long time (how long?).

The study also borders on being of local interest only, and it should probably be broadened a bit to make of interest to a wider audience.

Experimental design

The authors have done a good job of thinking through all the potential confounding factors (such as salinity change, food-web alterations, and habitat alterations) and demonstrating how each might cause unanticipated consequences.

The one thing that needs more attention is the “estuary signature” issue. What appears to be contradictory results are simply differences among estuaries because of antecedent conditions, long-term average estuary conditions, or residence times. An estuary where the salinity is low (<5 psu) does not change much when a lot of water is added, but and estuary where salinity is high (>25) changes a lot when water is added. Also the relative volumes of the water flow and the receiving waters is a critical metric. A bay that is small and water age is short, responds very differently from a bay that is large and the water age is long. This means that you have to be careful not to make broad sweeping conclusions, that might in fact only reflect a specific type of system.

Validity of the findings

The paper is unique and tackles a problem that is not often studied, so I think it is important. Also, there will be a lot of focus on restoring Gulf of Mexico habitats over the next 30 years because of the restore funding made available by the Deepwater Horizon settlement, so the review is timely.

While I agree that the literature reviewed is appropriate, I think there is a lot more literature from Mobile Bay, AL, and the Sacramento and San Joaquin River Delta area in CA that are relevant. This is a tough one because no review that is manuscript in length can cover everything, I do think there are some lessons from CA that are useful because the flow volumes there are also large like the MS River volumes.

The findings and conclusions are sound, but I do think that the finding should be tempered by what I call the estuary-signature issue (described above and below).

Additional comments

The reference list and citations within the text need a thorough review, there are many inconsistencies. And, there needs to be a check for missing references. Some specific comments are below:

L32: “results … are optimistic” doesn’t make sense. Plans can be optimistic. Better revise this sentence.
L34: Abstracts are better when written in the third person.
L41-44: An abstract should end with the significance of the findings. While I agree with the sentiments expressed in the last two sentences, they are not conclusions of this study.
L106: there is a ton of benthic stuff published on Mobile Bay too, which should be relevant to a review, particularly by Gary Gaston, and Chet Rackocinski. There is a ton of work on meiofauna from John Fleeger and his students in the LA bays and marshes.
L195: I think this reference should be Palmer et al. or Palmer, Montagna, and Kalke.
L220, 222, 252, 258, and many more: VanDiggelen and Montagna?
L400-402: Needs a little tweaking. While it is true that systems with high flow volumes (like the MS River) wash Chl downstream, systems with lower flow volumes (like some of the TX estuaries) can stimulate Chl, and you see positive correlation with flow. Point is that you can get different responses in different kinds of systems depending on the overall volumes of flow and the receiving bay. Thus a better indicator than flow rate is turn-over or residence time of the water.
L477: Interesting section, but stuff about seagrass response is a bit surprising. Which seagrass species increase in which salinity range? Most seagrass species prefer high salinity and marine conditions, and high flow rates decrease seagrass cover. One exception is Vallisneria, which likes salinities in the range of 5 psu or so. There is a lot of literature from the west coast of FL on this, particularly by Peter Doering.
L554: Again, it depends on the system and antecedent conditions. In estuaries where the average salinity is low (~ 10-15) then floods kill oysters because salinities go near zero. But in estuaries where average salinity is high (~25-30) then floods enhance oysters because salinities go to their optimal range around 18. There are tons of papers on this from the Gulf, particularly Apalachicola Bay in FL, Copano, San Antonio, and Galveston Bays in TX, and Breton Sound, LA. BTW, this explains why there is no conflict between Turner (Estuaries and Coasts 29:345-352, 2006) and Buzan et al. (Estuaries and Coasts 32:206-212, 2009). You would be doing the world a favor by pointing this out.
L566: I think this is the same story as above. Just be careful here.
L604: Do you mean “… response to sediment flux caused by freshwater flow diversions..”? Or the other way around “…response to salinity change caused by freshwater flow diversion to deliver sediment to the…”?
L611-615: Again, I think it is misleading to make blanket statements without considering antecedent conditions, or a change in the average conditions.
L641: replace since with because.
L675: can you provide a link to this report?
L 691: out of order.
L752: caps?
L765: incomplete reference
L771: caps?
L813-814: punctuation

·

Basic reporting

The review has a clear and important purpose, will be of interest to many researchers, and is appropriate for this journal.

I think the introduction needs some work:
1. I suggest reorganizing to lay out all of the expected effects of the diversions at the beginning then discuss the potential impacts of each in the sections. As written, the paper seems very long and it’s not clear what is coming next. There are nice transition sentences between sections, but I think it would be better to lay out the outline clearly in a bit more detail at the beginning and remove or shorten the transitions.
2. The explanation of the diversions in the system is confusing and needs clarification and some more detail. I suggest modifying figure 1 to specify where the water goes from each of these diversions with arrows and also include information about how much water is diverted at each one (maybe visualized as arrow size in the figure). I’m not totally clear on the difference between the existing “freshwater” diversions and proposed “sediment” diversions (line 66), more explanation here would be helpful.

Experimental design

The literature review seems fairly thorough to me, although I had a few minor points:
1. Lines 330-335 this is an oversimplification and not necessarily accurate. See classic work by Sanders in Buzzards Bay.
2. Line 409 ref needed here

The review is organized well into sections that make sense, but I recommend some reorganization to put more of the predicted effects of the diversions at the beginning and focus in the later sections on how these predicted effects will affect benthic communities (see above).

Figure 2 does not seem necessary.

Validity of the findings

The conclusions and points for future research seem good to me. The authors have compiled a lot of useful information and have written a nice synthesis, but I think the paper would really benefit from a well-conceived conceptual diagram to summarize these findings. Figure 3 is sort of confusing and not particularly informative, and the two tables contain an overwhelming amount of information. A really nice diagram that clearly conveys the major effects of freshwater diversions on benthic communities would be widely used in teaching and research (at least, I would like to have a good diagram on this for my teaching!). I recognize that developing really good conceptual diagrams is challenging, but I strongly encourage the authors to spend some time on this.

Additional comments

This manuscript is really interesting and generally well written but does seem a little long and somewhat wordy in places, I recommend editing to make it a little more concise.

Minor points:
Lines 269-271 awkward sentence
Lines 307-309 these numbers aren’t comparable, need to have the same units.

---

## Round 0.2 · Minor Revisions

There are only a few minor issues that need your attention. The reference associated with Line 176 as pointed out by reviewer #1 and the issue regarding Figure 3 as detailed by reviewer #2. Please attend to these or provide your detailed disposition regarding each.

Reviewer 1 ·

Basic reporting

The authors have done a good job of responding to all suggestions, and this section is now good.

Experimental design

The authors have done a good job of responding to all suggestions, and this section is now good.

Validity of the findings

The authors have done a good job of responding to all suggestions, and this section is now good.

Additional comments

L 176: Meiofauna in estuaries are usually defined as <0.5 mm and > 0.063 mm. There are hundreds of references on this, but using one of the books by Olav Giere is best.

·

Basic reporting

The authors have done a nice job of addressing my comments. Figure 1 is much better. I still have a few concerns about Figure 3 – most importantly, it has red and green arrows that will not be distinguishable by someone who is color-blind or even when printed on a black and white printer, the colors need to be changed and ideally distinguishable by line type as well as color. Why will increased sedimentation increase primary production? The figure doesn’t really match the text. Consider adding arrows for unknown effects – the text discusses potential but unknown effects on benthic community structure, diversity.

Experimental design

no comment

Validity of the findings

no comment

---

## Round 0.3 · accepted · Accept

All of the reviewer concerns have been sufficiently accommodated. This should be a well received mini-review.